# A lightweight YOLOv7 insulator defect detection algorithm based on DSC-SE

**Yulu Zhang**[1], **Jiazhao Li**[1], **Wei Fu**[1], **Juan Ma**[2,3], **Gang Wang**[1] *

**1** School of Electrical and Information Engineering, Beihua University, Jilin, China, **2** Chui Yang Liu Hospital, Tsinghua University, Beijing, China, **3** Affiliated Hospital of Beihua University, Jilin, China

* bhwanggang@163.com

**Data Availability Statement:** Data relevant to this study are available from github.com//InsulatorData/insulatorDataSet.git.

**Funding:** This research was funded by the scientific research project of Jilin Provincial

## Abstract

As the UAV(Unmanned Aerial Vehicle) carrying target detection algorithm in transmission line insulator inspection, we propose a lightweight YOLOv7 insulator defect detection algorithm for the problems of inferior insulator defect detection speed and high model complexity. Firstly, a lightweight DSC-SE module is designed using a DSC(Depthwise Separable Convolution) fused SE channel attention mechanism to substitute the SC(Standard Convolution) of the YOLOv7 backbone extraction network to decrease the number of parameters in the network as well as to strengthen the shallow network's ability to obtain information about target features. Then, in the feature fusion part, GSConv(Grid Sensitive Convolution) is used instead of standard convolution to further lessen the number of parameters and the computational effort of the network. EIoU-loss(Efficient-IoU) is performed in the prediction head part to make the model converge faster. According to the experimental results, the recognition accuracy rate of the improved model is 95.2%, with a model size of 7.9M. Compared with YOLOv7, the GFLOPs are reduced by 54.5%, the model size is compressed by 37.8%, and the accuracy is improved by 4.9%. The single image detection time on the Jetson Nano is 105ms and the capture rate is 13FPS. With guaranteed accuracy and detection speed, it meets the demands of real-time detection.

## 1 Introduce

As power grid systems continue to improve, the length of overhead transmission lines is increasing, and insulator strings are crucial components in these lines. They provide mechanical stability and electrical insulation to support the conductor and prevent it from touching the tower. However, long-term exposure to environmental factors can cause insulators to break, self-destruct, or develop defects. This can have a significant impact on the reliability and safety of power grids, which in turn can pose a risk to society's economy. Therefore, detecting flaws in insulators is essential for the operation and maintenance of electricity grids. Insulator detection is usually carried out by professional inspectors who have extensive experience in recording the operating condition of insulators. The limitations of the distribution range of overhead transmission lines and the overhead environment make it difficult to inspect insulators for minor surface defects through manual observation with the naked eye or binoculars. Climbing

Science and Technology Program(20190303038SF and 20200404154YY). The funders had no role in study design, data collection and analysis, decision to publish, or preparation of the manuscript.

**Competing interests:** The authors declare no conflict of interest.

up the tower for a closer look is an inefficient method, making it challenging to meet the requirements of real-time insulator inspection with manual inspection alone. As a result, using UAVs to take photos and carry out damage identification has become a mainstream inspection method, which greatly reduces the maintenance workload of field staff [1, 2].

Traditional methods for insulator image detection often rely on manual intervention for feature extraction, for example, using Hough transform detection, watershed algorithms, edge detection, and utilization of information on space and color [3–7]. These techniques have limited improvement potential, are weakly generalizable, and make it difficult to satisfy the demands of small target identification in complex backdrops. They are commonly deployed for large-scale target images featuring simple backgrounds.

As computer technology develops rapidly, it also drives the rapid development of deep learning. Deep learning is a neural network-based artificial intelligence strategy that enables the automatic classification and recognition of data through the training and optimization of large amounts of data. Deep learning has become a popular tool for power system target detection in recent years because it can swiftly and accurately locate targets in images [8–10]. Convolutional neural networks are commonly used in deep learning and can be used to detect defects in transmission lines, substations, and other equipment to reduce the workload of manual inspections. The difficulty and workload of feature extraction are significantly reduced by deep learning algorithms' ability to automatically learn features from the original data without human extraction of features. Furthermore, the performance of deep learning-based object detection is excellent, and the methods can be split into two main categories. One category includes two-stage algorithms like Mask R-CNN [11], Fast R-CNN [12], Faster R-CNN [13], and Feature Pyramid Network (FPN) [14], which generate candidate frames for possible targets before identifying the target class and correcting the candidate boxes. Another class is the one-stage target detecting algorithm, which is represented as the YOLO family [15–18] and SSD [19]. This algorithm can pinpoint the target's location on the picture and forecast the category's confidence level. Because of their quick detection times, compact models, and adaptable deployment, single-segment YOLO series algorithms are frequently used in the industry. They can perform both target recognition and boundary regression.

For target detection, Zhao et al. [20] used a fine-tuning method to improve the anchor frame generation method of Faster R-CNN and non-extreme suppressing in the area of suggestion networking to solve the detection problem of mutual occlusion. Tan et al. [21] suggested a method to improve the R-CNN network by fusing gradient, texture, and grayscale features to improve recall and accuracy. Li et al. [22] proposed an improved YOLOv5 insulator detection model with light correction enhancement, which converted the image from RGB color space to HSV space and corrected it by a two-dimensional adaptive gamma transform to improve the detection accuracy of insulators under light interference. According to Feng et al. [23], a YOLOv5-based automatic insulator detection method with K-value clustering successfully locates and identifies insulator flaws, but the detection accuracy is only 86.8%. The aforementioned techniques have improved the accuracy of detection to a satisfactory level, but they are still time- and resource-intensive to compute. To achieve lightweight processing, Wu et al. [24] presented a Centeret-based insulator inspection approach that reduced redundant information and enhanced network detection accuracy while also simplifying the backbone network to achieve lightweight processing. Sadykova et al. [25] provided a cost-effective method for aerial insulator detection by UAV based on YOLO's deep learning neural network model, which improved the insulator detection efficiency. Zhang et al. [26] proposed a SOD-YOLO model based on UAV image analysis, using the K-means algorithm to re-cluster, using channel pruning to lighten the processing, and adding a CBAM attention mechanism to improve the detection accuracy of small targets. Qiu et al. [27] compared YOLO-based UAV real-time

detection algorithms, and the experimental results showed that YOLOv2 and YOLOv4-Tiny based on resnet50 have better detection accuracy and speed. Tulbure et al. [28] talked about modern defect detection models based on deep convolutional neural networks to provide a more efficient method for modern industrial inspection tasks. Cao et al. [29] used GhostNet to lighten the network, which reduced the memory consumption but the model's robustness was subpar. Yang et al. [30] proposed an improved YOLOv3 algorithm for UAV detection of insulators, using EIoU as a regression loss function, which significantly improved the overlap between the prediction frame and the real frame and accelerated the convergence rate.

Unmanned aircraft systems (UAS) are equipped with limited computing memory and resources, which poses a challenge to the high computational complexity of embedded models. This complexity is divided into spatial and temporal components, with spatial complexity determining the number of parameters in the model and temporal complexity dictating detection time. However, many deep learning-based insulator detection techniques suffer from high computational complexity, slow detection speed, and difficulty in embedding them in UAV devices. In 2022, Wang et al. [31] proposed YOLOv7 as the latest target detection model, which surpasses all current models in terms of both detection speed and accuracy. YOLOv7 performs exceptionally well in detecting objects in images, providing precise location and classification information. YOLOv7-tiny is a lighter version of YOLOv7, with fewer parameters and lower computational complexity, making it suitable for real-time object detection in resource-constrained environments. However, YOLOv7-tiny may not be suitable for detecting tiny insulator defects, as it lacks sufficient salient features and does not meet the model requirements for size and contrast, resulting in reduced detection accuracy compared to larger and more complex models. There is not much research on applying YOLOv7-Tiny in UAS to inspect insulators, and there is room to improve the accuracy and detection speed of the model for the real-time nature of UAVs in insulator defect detection tasks. We provide a simple, quick, and effective insulator defect identification approach based on the YOLOv7-tiny model to address this issue. The primary contributions are as follows: The DSC-SE module utilizes the Depthwise Separable Convolution and SE attention mechanism as a small parametric and enhanced feature extraction module to replace the normal convolution of the YOLOv7-tiny backbone feature extraction network. This reduces the size of the model and enhances the detection of small targets. The feature fusion network uses GSConv, which has comparable extraction capability to standard convolution, to reduce the time complexity of the model. Additionally, the VOVGSCSP module is designed with residual edges based on the GSConv to accelerate inference while maintaining accuracy. Finally, the EIoU loss is chosen as the localization loss function to accelerate the convergence of the model and to better measure the similarity between the detection results and the real insulator frame.

## 2 Related algorithms

### 2.1 YOLOv7

YOLOv7 is an upgraded version of YOLOv5, which introduces ELAN structure and MP (Max-Pool) structure based on YOLOv5's backbone extraction network with FPS in the range of 5–160. As shown in Fig 1A, the ELAN module is an efficient network structure with two branches. One goes through a 1×1 convolution to do the channel number change, the other first goes through a 1×1 convolution module to do the channel number change and then goes through four 3×3 convolution modules to do the feature extraction, and finally, the two branches are summed to get the final feature extraction result. The ELAN module continuously enhances the learning capability of the network by controlling the shortest and longest gradient path. As shown in Fig 1B, the MP module also has two branches that serve to perform

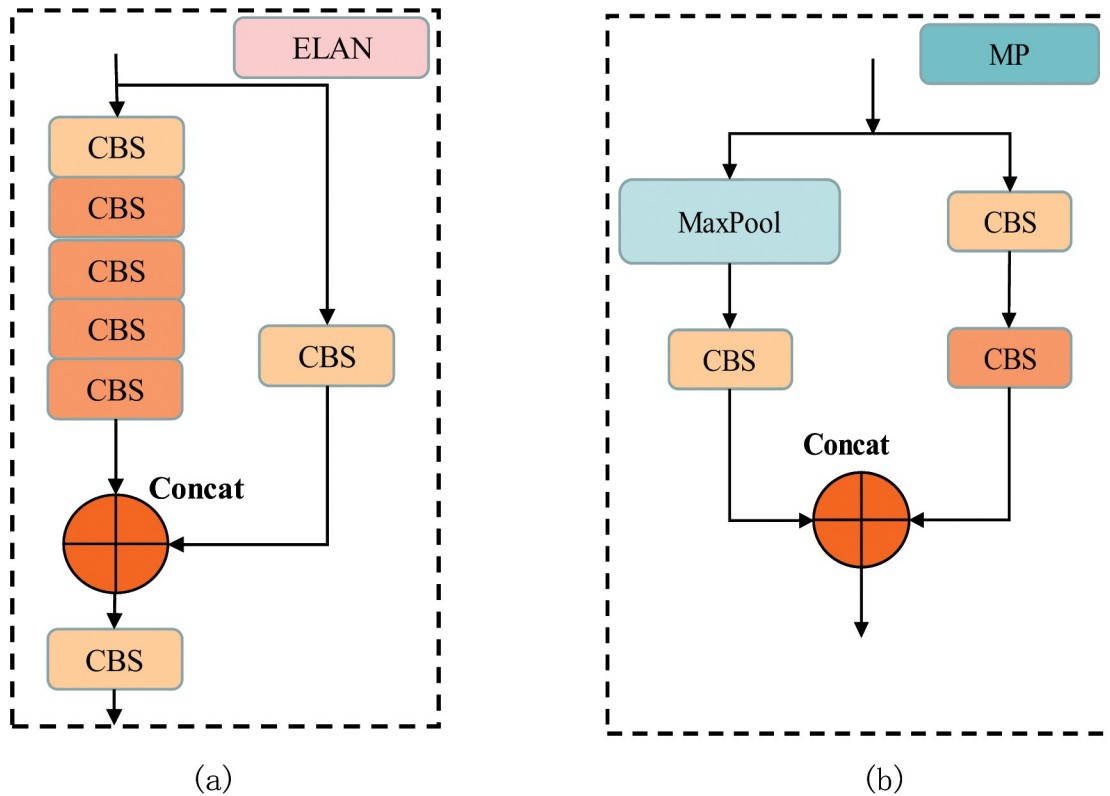

**Fig 1. Improved part of YOLOv7.** (a) shows the ELAN network structure; (b) shows the MP network structure.

downsampling. The first branch involves a maximum pooling layer for downsampling followed by a 1×1 convolution to change the channel count. The second branch, on the other hand, first undergoes a 1×1 convolution to change the channel count and then a 3×3 convolution kernel with a 2-step convolution block, which is also downsampled. The results of the two branches are added together to get the result of super downsampling.

In addition, YOLOv7 has a low computational cost and efficient training speed, which makes it promising for a wide range of practical applications. A more compact variant of YOLOv7, YOLOv7-tiny, has a quicker detection rate and less computational complexity. The model has a good performance-to-speed ratio and is appropriate for real-time object detection in resource-constrained environments. The backbone network, head network, and output layer are the three divisions of the YOLOv7-tiny network. ELAN and MP structures are used in the network's backbone for feature extraction. where the ELAN structure is stacked with 2 fewer convolutional layers than the standard version of YOLOv7, the activation function becomes LeakyReLU, which does not change the width and height of the input feature layers, and enhances the interaction between each feature layer through expansion, random combination and splicing to enhance the learning capability of the network; The head network contains several convolutional layers and layers of pooling for additional feature extraction. The output layer, which is the core component of YOLOv7-tiny, has detection and classification heads for identifying and categorizing objects. Mosaic data augmentation is a new training method used by YOLOv7. Multiple images can be stitched together to form a larger image using mosaic data augmentation, increasing the diversity and amount of training data. This method can improve the model's accuracy and generalization.

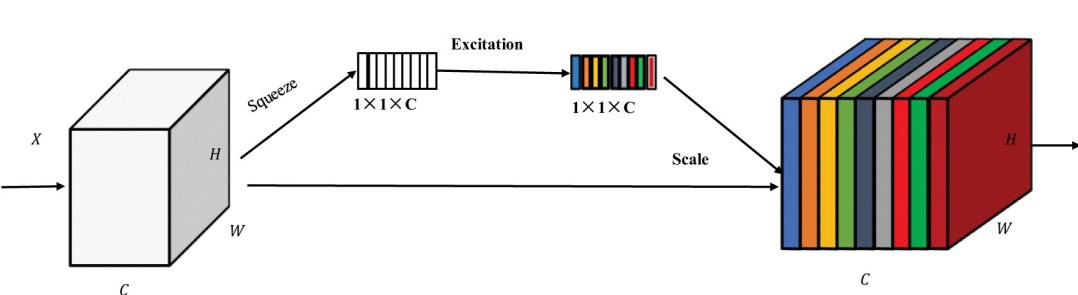

**Fig 2. SE network architecture.**

## 2.2 SE attention

The SE (Squeeze-and-Excitation) [32] attention mechanism is a convolutional neural network attention mechanism, as shown in Fig 2. Its main idea is to improve model performance by assigning channel weights adaptively. Following the convolutional layer, the SE attention mechanism obtains the average response of each channel via a global pooling layer before generating the weight coefficients for each channel via two layers that are fully interconnected (a compression layer and an activation layer).

To improve the model's attention to important features, these weight coefficients are used to weigh each channel in the feature map. The following benefits pertain to the SE attention mechanism:

1. It improves the model's performance by adaptively learning the significance of each channel.

2. Global pooling and fully connected layers are used to implement it; no additional parameters are needed.

3. It may be included in already-existing convolutional neural networks, enhancing model performance without requiring complete model retraining.

The SE module enhances feature representation by adaptively learning the relationships between feature channels. In insulator detection, the SE module helps the network to better understand the insulator features and improve the focus on critical information. This helps to improve the accuracy and robustness of insulator detectors.

## 3 Improved YOLOv7-tiny network structure

### 3.1 DSC-SE

The YOLOv7-tiny backbone network utilizes a multitude of convolutional blocks that comprise 3x3 standard convolutional layers and pooling layers in a series to extract features. The input image undergoes processing by the backbone network to generate a sequence of feature maps, as illustrated in Fig 3. However, the shallow layer network has a small perceptual field, and each neuron can only capture local information from the input image. The shallow network has fewer layers and a limited number of filters that extract the basic features of insulator texture and color. This makes the network insufficient in extracting insulator pixels with small defects and a large number of parameters, making it challenging to extract maximum effective feature information with smaller arithmetic power.

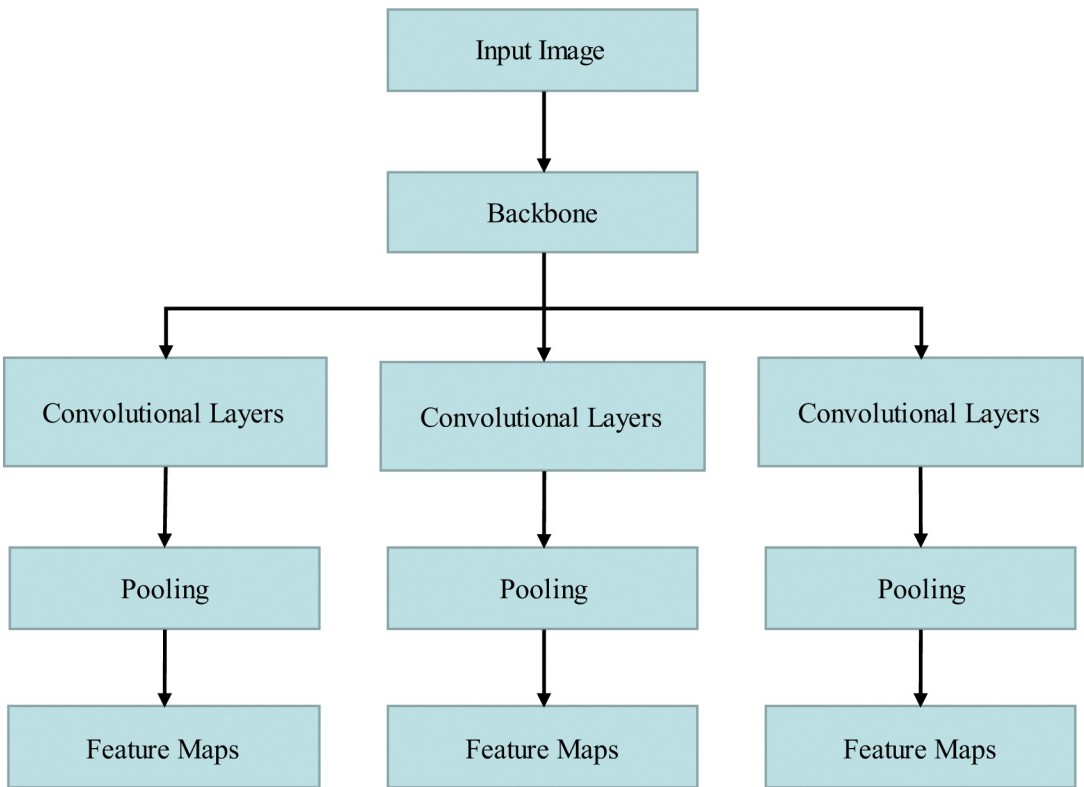

**Fig 3. Schematic diagram of the convolutional extraction information of the backbone network.**

To solve the above problems, a lightweight module DSC-SE based on deep convolution and attention mechanism is designed, and the structure is shown in Fig 4.

DSC-SE module through a 1×1 convolution BN processing and Hard-swish activation function after and multiple 3×3 DSC in series and through the SE module of channel attention mechanism, can solve the problem of extracting too much interference information, and then spliced with 1×1 convolution to achieve feature extraction, it can get more important feature information to some extent and enhance the model's ability to obtain features of insulators and defective parts.

Defects in insulator inspection tasks often have small dimensions and low contrast, which makes accurate detection a challenge. In this case, the DSC depth-separable convolution module plays an important role. By decomposing the standard convolution into depth and point-by-point convolutions, the DSC module reduces computational complexity while maintaining expressiveness. This is very beneficial for insulator detection because the task requires real-time or quasi-real-time analysis over a large number of image samples, so

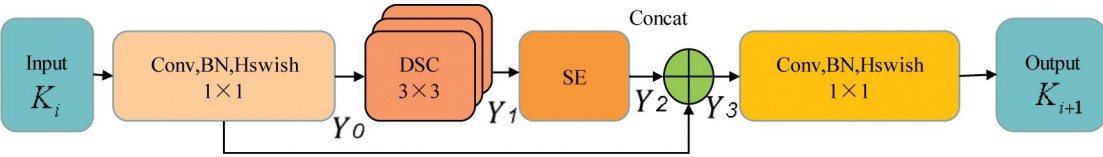

**Fig 4. Structure diagram of the DSC-SE module.**

efficiency is critical. Assuming that there are M input channels for standard convolution, N output channels, $D_k \times D_k$ size for the convolution kernel, and H×W size for the output characteristic graph, the ratio of the standard convolution to the Depthwise Separable Convolution is $R_Q$:

$$R_Q = \frac{H \times W \times D_k \times D_k \times M + H \times W \times M \times N \times 1 \times 1}{H \times W \times D_k \times D_k \times M \times N} = \frac{1}{N} + \frac{1}{D_k^2} \tag{1}$$

From Eq (1), it can be seen that the number of parameters is about 1/9 of the standard convolution after processing with 3 × 3 DSC, which reduces the number of parameters while ensuring the extraction of more feature information. The depth-separable convolution is used in layers 3 and 5 of the backbone network, and the DSC-SE module is used in layers 2, 7, 8, 9, 10, 11, 12, 13, and 14. The calculation process of the DCS-SE module is:

$$Y_0 = Conv(K_i) \tag{2}$$

$$Y_1 = Concat(Dw_d(\ldots Dw_2(Dw_1(Y_0))), Y_0) \tag{3}$$

$$Y_2 = SE(Y_1) \tag{4}$$

$$Y_3 = Conact(Y_0, Y_2) \tag{5}$$

$$K_{i+1} = Y_0(Y_3) \tag{6}$$

where, denotes the feature map extracted by standard convolution for the input (taking 1, 2...n), denotes the result of deep separable convolution, denotes the feature map obtained by SE attention mechanism, denotes the result of splicing with, and denotes the final extracted feature map of DSC-SE obtained by standard convolution.

The DSC-SE module, while globally extracting the channel information, adaptively learns the dependencies between different channels with the help of SE, adaptively enhances the feature pixels of the target, weakening the interference of complex background information on insulator fault detection and ensuring the accurate extraction of defects by the network while expanding the shallow network perception field.

## 3.2 GSConv convolution and VOVGSCSP module

To address the problem that insulators are complex and small targets in the overhead scene, and the depth-separable convolution extraction will lose a large amount of channel information, we introduce the lightweight convolution GSConv—which is a hybrid convolution of SC, DSC, and shuffle [33]. The shuffle operation, as shown in Fig 5 below, is used to permeate the information generated by SC into each part of the information generated by DSC. By exchanging local feature information uniformly on different channels, this method allows the information from the normal convolution to be completely mixed into the output of the deeply separable convolution, which can reduce the computational cost while maintaining maximum accuracy.

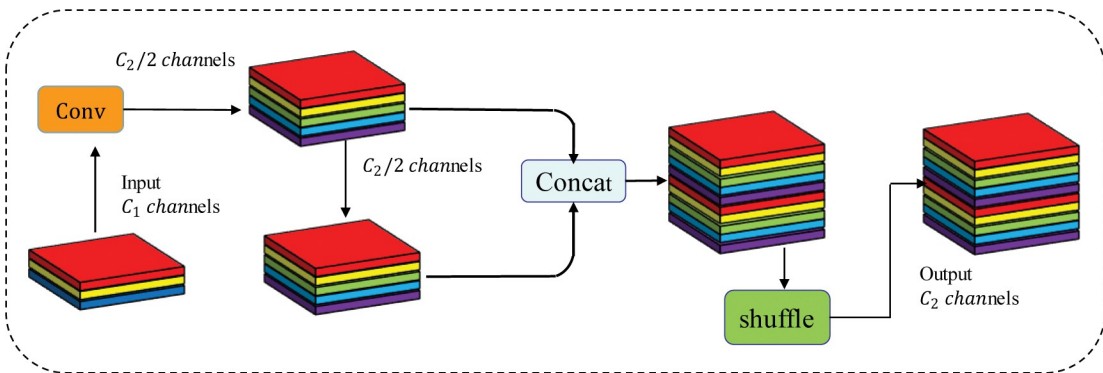

**Fig 5. The network architecture of GSConv.**

The FLOPs are typically used to describe the convolutional computation's time complexity. As a result, the SC, DSC, and GSConv time complexity is:

$$Times_{SC} = (W \times H \times K_1 \times K_2 \times C_1 \times C_2) \tag{7}$$

$$Times_{DSC} = (W \times H \times K_1 \times K_2 \times C_2) \tag{8}$$

$$Times_{GSConv} = [W \times H \times K_1 \times K_2 \times \frac{C_2}{2} \times (C_1 + 1)] \tag{9}$$

where W is the original feature map's width and H is the output feature map's height. The number of input channels per convolution kernel is $C_1$, the number of channels in the output feature map is $C_2$, and $K_1 \times K_2$ is the magnitude of the convolution kernel. It can be seen that the time complexity of GSConv is less than that of SC and DSC. In insulator detection, the GSConv module enables the network to focus more on the grid region where the insulators are located and extract more accurate features by adaptive convolution. This helps to improve the localization accuracy and detection performance of the insulator detector.

This paper proposes the use of lightweight convolutional GSConv instead of SC to reduce the computational cost by up to 50%. However, a new model is still needed to further reduce inference time while maintaining accuracy. To address this, the paper introduces the VoV-GSCSP network module, shown in Fig 6, which is based on the GSConv primary aggregation method and draws inspiration from VoVNet [34] and CSPNet [35]. The module replaces ordinary convolution with GSConv in the feature fusion network and replaces five

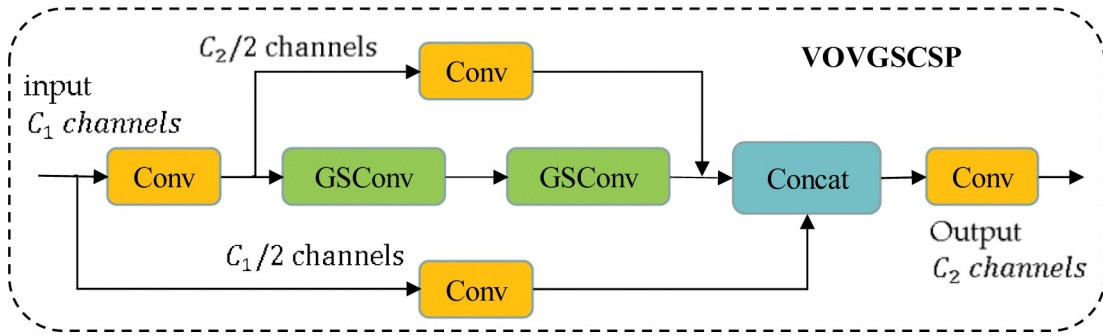

**Fig 6. The network architecture of VOVGSCSP.**

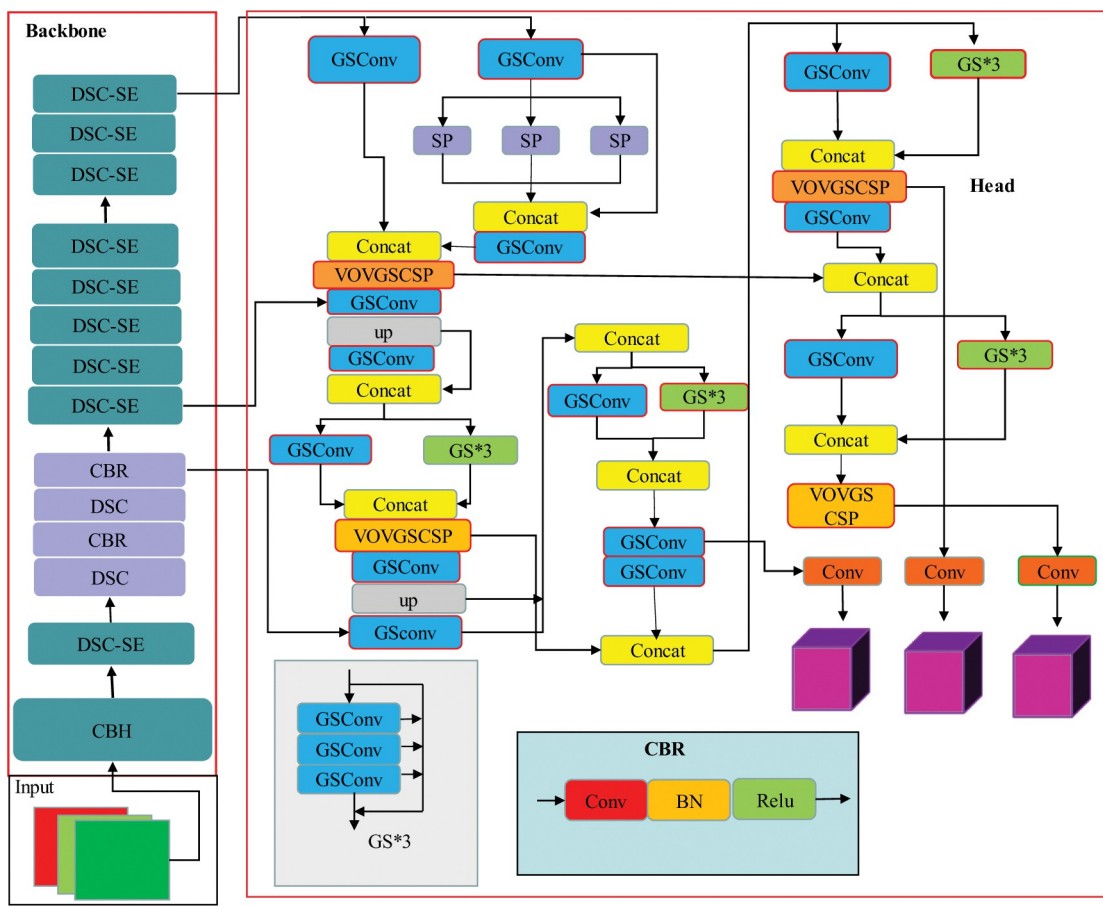

**Fig 7. The improved network structure of YOLOv7-tiny.**

ordinary convolutions after the Concat layer with the VoV-GSCSP module. This approach speeds up computation and inference while ensuring accuracy.

The framework of the light-weight insulator defect inspection model based on the refined YOLOv7-tiny is shown in Fig 7 below by replacing the standard convolution of the backbone extracting features network with the depth-separable convolution and DSC-SE modules, and by introducing the GSConv convolution and VOVGSCSP modules in the feature fusion network.

## 3.3 Improvement of the loss function

In the YOLOv7-tiny model, classification loss, localization loss, and confidence loss are weighted sums that add up to an overall net loss. The confidence loss and categorization loss functions among them use the binary cross-entropy loss, and the localization loss function uses the CIoU-loss.

$$L_{CIoU} = 1 - IoU + \frac{\rho^2(b, b^{gt})}{s^2} + \alpha v \tag{10}$$

$$\alpha = \frac{v}{(1 - IoU) + v} \tag{11}$$

$$v = \frac{4}{\pi^2} \left( \arctan \frac{w^{gt}}{h^{gt}} - \arctan \frac{w}{h} \right)^2 \qquad (12)$$

The formula for calculating the Euclidean distance (ρ) between two locations is given by the centroids (b and c) of the predicted and actual boxes. The diagonal length of the minimal bounding rectangle of the predicted and actual boxes is represented by s, while the aspect ratio similarity of the two frames is represented by v. The dimensions of the actual box are denoted by a and b, while the width and height of the anticipated box are represented by w and h, respectively. The ratio's parameter is used to balance the equation.

Although the CIoU loss considers the boundaries of the regression's aspect ratio, the centroid of distance, and overlapping region. However, returning to the term v in Eq(12), it can be seen that there are two problems: (1) uses the aspect ratio of the prediction box and the target box, then when the prediction box's size satisfies $\{(w = kw^{gt}, h = kh^{gt}) | k \in R^+\}$, the penalty term of this item in CIoU loses its effect. (2) According to the following gradient formula of w, h, it can be concluded that $\frac{\partial v}{\partial w} = -\frac{h}{w} \frac{\partial v}{\partial h}$ are inversely related to each other. That is, when one of w, h increases during training, the other must decreases, which occasionally inhibits the model from efficiently optimizing the similarity.

$$\frac{\partial v}{\partial w} = \frac{8}{\pi^2} \left( arctan \frac{w^{gt}}{h^{gt}} - arctan \frac{w}{h} \right) \frac{h}{w^2 + h^2} \qquad (13)$$

$$\frac{\partial v}{\partial h} = \frac{8}{\pi^2} \left( arctan \frac{w^{gt}}{h^{gt}} - arctan \frac{w}{h} \right) \frac{w}{w^2 + h^2} \qquad (14)$$

EIoU Loss [36] is selected as the network's regression loss function in this paper to address this issue. The overlap loss $L_{Iou}$ between the forecast box and the real box, the central distance loss $L_{dis}$ between the prediction frame and the real frame, and the width and height loss $L_{asp}$ between the prediction frame and the real frame are the three components that EIoU divides the loss function into.

$$L_{EIoU} = 1 - IoU + \frac{\rho^2(b, b^{gt})}{s^2} + \frac{\rho^2(w, w^{gt})}{C_w^2} + \frac{\rho^2(h, h^{gt})}{C_h^2} \qquad (15)$$

where $C_w^2$ and $C_h^2$ are the width and height of the minimal bounding rectangles of the predicted and real frames, respectively.

The first two parts of the EIoU loss follow the CIoU method, and the width-height loss is such that the difference between the width and height of the predicted and real boxes is minimized, resulting in faster convergence. In insulator detection, the EIoU loss function can better measure the similarity between detection results and real insulator frames, thus improving the learning of the model and enhancing the detection accuracy.

## 4 Experiments

### 4.1 Data set introduction

The experimental dataset used in this study comprises two parts: the China Academy of Electric Power (CPLID) dataset from GitHub and the Baidu Internet images dataset. The former includes 1417 images of insulators, out of which 600 are normal and 647 are broken. The images were captured via aerial photography of UAV inspection sites. The dataset was annotated using the open-source LabelImg annotation tool and categorized as insulator and defect.

The completed dataset was randomly divided into training, validation, and test sets in the ratio of 7:2:1 to ensure an unbiased evaluation of the model. Given the need to reduce overfitting during training, a large number of data samples were used for model training. The training set was augmented using various techniques such as mirroring, flipping, changing contrast, and saturation resulting in a total of 1732 samples.

## 4.2 Evaluation index

Precision P, recall R, and mAP(mean average precision) are common assessment metrics that were utilized in this experiment to evaluate the model's performance. The formulas are as follows:

$$P = \frac{TP}{TP + FP} \tag{16}$$

$$R = \frac{TP}{TP + FN} \tag{17}$$

$$AP = \int_0^1 P(r)dr \tag{18}$$

$$mAP = \frac{1}{n}\sum_{i=0}^{n} AP_i \tag{19}$$

The positive samples correctly identified by the model are referred to as TP, while negative samples incorrectly identified are referred to as FP. Misclassification of positive samples as negative classes is denoted by FN, where n represents the number of sample classes in the data set. Furthermore, the number of model parameters, floating point operations, and detection speed metrics were increased to provide a more accurate estimation of the performance of the lightweight model.

## 4.3 Experimental environment and configuration

Table 1 shows a portion of the experimental setting used in this paper, which employed the Pytorch framework in a Python 3.9 environment to implement the full algorithm of the model.

The paper specifies that the experimental training process was conducted using the gradient descent method with the Adam optimizer. The initial learning rate was set to 0.01 and the learning rate was adjusted using Cosine Annealing LR. The image input size was 640 × 640 and the batch size was 16. The total training comprised 150 rounds.

Table 1. Configuration of the experimental environment.

| Configuration | Version |
|---|---|
| CPU | AMD 5600H |
| GPU | Nvidia GeForce GTX 3050 |
| CUDA | 11.6 |
| Operation system | Windows11 |
| Frameworks | Pytorch |
| Compilation environment | Pycharm |

**Table 2. Performance comparison of the different loss functions.**

| Loss Function | Precision(%) | Recall(%) | mAP0.5(%) |
|---|---|---|---|
| CIoU | 92.4 | 86.3 | 93.4 |
| SIoU | 93.8 | 87.3 | 93.3 |
| **EIoU** | **95.2** | **87.6** | **93.8** |

## 5 Experimental results and analysis

YOLOv7 originally employed CIoU as the localization loss function. To choose a more suitable loss function, we chose the performance of EIoU and SIoU on the YOLOv7 model to compare with it. The comparative effects are illustrated in Table 2.

As indicated in Table 2 above, the usage of EIoU can increase precision and recall by 2.8% and 1.3% in comparison to the other two approaches. The outcomes of the experiment demonstrate that the EIoU loss function performs well in insulator fault identification.

In Fig 8, we present the PR curves of our improved model, where the vertical axis represents the precision rate P and the horizontal axis represents the recall rate R. The PR curves indicate that the accuracy rate only slightly decreases as the recall rate increases. When the confidence level is set to 0.5, the mean average precision (mAP) for detecting normal insulators is 92.5%, for defective insulators is 95%, and for the total category is 93.8%.

To evaluate the detection performance of the algorithm, we compared the improved algorithm with the commonly used algorithm for detecting insulators in terms of accuracy rate and loss value. The accuracy curves are presented in Fig 9A, where the average accuracy of our improved model remains stable at 95% after 100 epochs, while YOLOv7-tiny remains stable at 90%. The loss curves of different models are shown in Fig 9B, where it can be observed that the loss curves of different models decrease rapidly within about 30 epochs and eventually stabilize

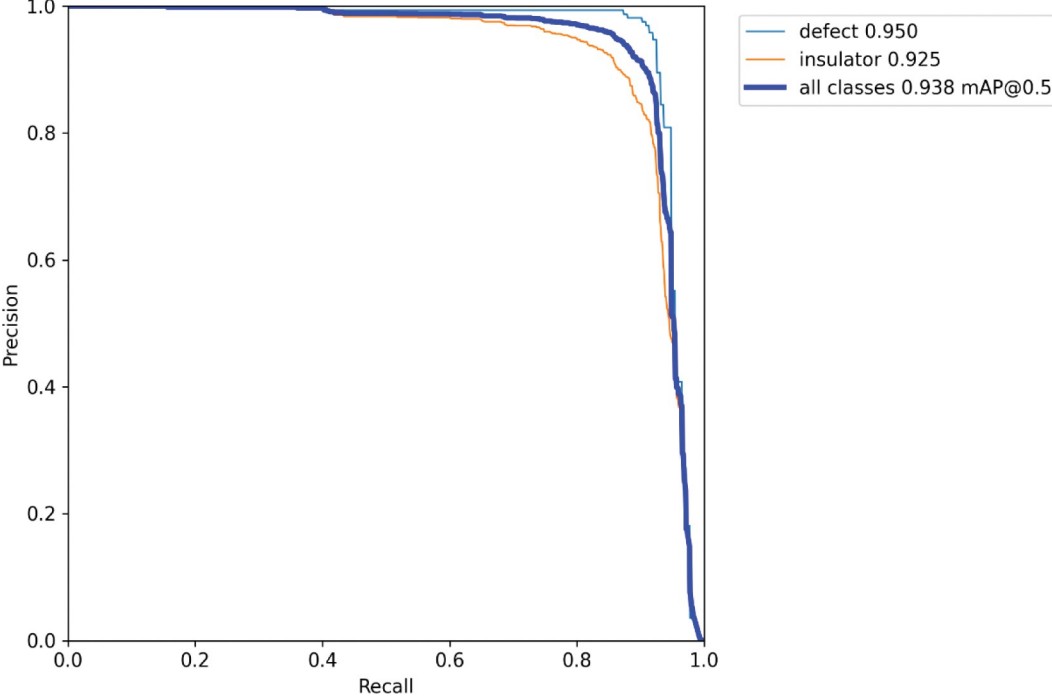

**Fig 8. The precision-recall curve of our model.**

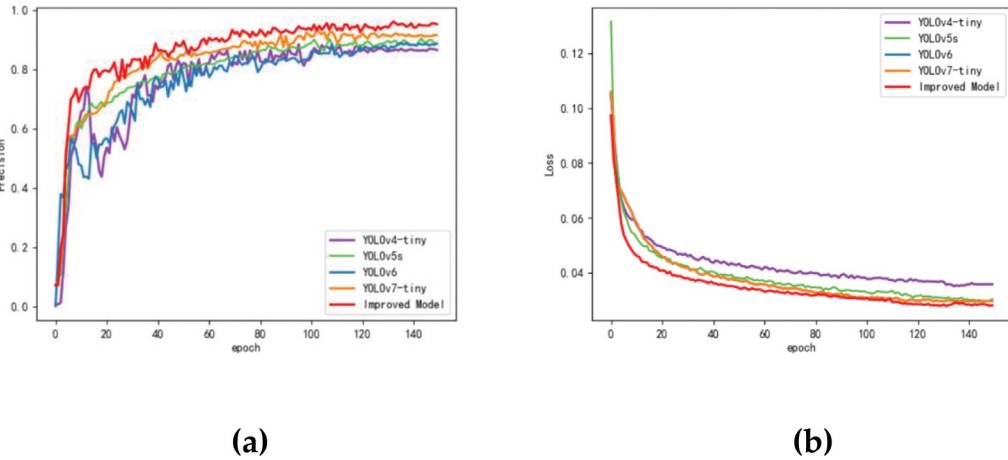

**(a)** **(b)**

**Fig 9. Comparison of different methods.** (a) Precision Curve; (b) Loss Curve.

after 100 rounds. Our improved model demonstrates a faster decline and better convergence than YOLOv7-tiny within 20 rounds, indicating that the adjustment of the loss function improves the network's convergence. YOLOv4-tiny exhibits the worst loss.

### 5.1 Ablation test

Several ablation experiments were created for comparative verification to test the performance of the improved module. The results are listed in Table 3, $\sqrt{}$ which shows the addition of this module.

The table displays the detection findings of the original YOLOv7 model in the first row. The introduction of the DSC-SE module into the backbone extraction network results in a decrease of 30.4% in parameters and 47.8% in computational effort. However, the mAP value decreases due to the reduction of the improved model parameters and the number of convolutional layers. In the feature fusion part, the use of GSConv and VOVGSCSP modules instead of the standard convolution leads to a 13% and 11% decrease in calculation and parameter quantities, respectively. The mAP improves by 0.5% when using EIoU-loss as a localization loss without additional network size and complexity. The VOVGSCSP module results in a 0.7% increase in mAP for the entire model, but requires a 5% and 10% increase in complexity and computational effort, respectively. However, the overall improved model mAP reaches 93.8%, with a 37.8% reduction in the number of parameters and a 54.5% reduction in computational effort. This makes it more suitable for edge terminals with limited hardware configuration, small size, and low power consumption. Taking into account the trade-off between

**Table 3. Results of ablation experiments.**

| Method | Improvement | | | | mAP(%) | GFLOPs (%) | Params(M) |
|---|---|---|---|---|---|---|---|
| | DSC-SE | GSConv | VOVGSCSP | EIoU | | | |
| 1 | | | | | 93.1 | 13.2 | 6.01 |
| 2 | √ | | | | 90.1 | 6.9 | 4.18 |
| 3 | √ | √ | | | 92.8 | 5.7 | 3.35 |
| 4 | √ | √ | √ | | 93.4 | 6.0 | 3.74 |
| 5 | √ | √ | | √ | 93.1 | 5.7 | 3.35 |
| 6 | √ | √ | √ | √ | **93.8** | **6.0** | **3.74** |

**Table 4. Performance comparison with other models.**

| Models | Precision(%) | Recall(%) | GFLOPs(G) | FPS |
|---|---|---|---|---|
| YOLOv4-tiny | 86.1 | 81.4 | 3.43 | 63 |
| YOLOv5s | 89.8 | 86.2 | 15.8 | 46 |
| YOLOv6 | 88.3 | 85.2 | 44.2 | 42 |
| YOLOv7-tiny | 90.3 | 92.4 | 13.2 | 50 |
| Our Model | **95.2** | **87.6** | **5.9** | **52** |

accuracy and number of parameters, the current improved model has advantages over other models.

## 5.2 Comparison of detection performance with other models

In order to determine the most suitable network model, we conducted a comprehensive experimental comparison of various common detection models from other architectures. As can be seen from Table 4, our improved model achieved the highest accuracy of 95.2% and ranked second in FPS, while also having significantly less computational complexity than YOLOv5s, YOLOv6, and YOLOv7-tiny. Although the YOLOv4-tiny model performs better in detecting insulators at a faster speed, it is less effective in identifying small defective insulators. The decrease in the recall rate compared to the original algorithm is attributed to the reduction of network parameters. Based on a comprehensive comparison, our proposed algorithm proves the effectiveness of the improved algorithm.

This paper explores the effectiveness of lightweight defect detection models, specifically focusing on YOLOv7-tiny. However, it is worth noting that this model may struggle with detecting very small insulator defects due to its size in comparison to larger and more complex models. Insulators are considered small target defects as they are caused by surface defects such as erosion, cracking, broken insulator skirts, and exposed mandrels. It is possible that very small defects may not have sufficient salient features or fail to meet the model's requirements for size and contrast, which can result in reduced detection accuracy. To evaluate the detection performance of the improved model, a comparison test was conducted on the UAV-based aerial photography test set. Fig 10 displays four different images from left to right, and the results obtained using YOLOv4-tiny, YOLOv5s, YOLOv6, YOLOv7-tiny, and our improved method detection are shown from top to bottom.

Fig 10A shows the case with low light, and Fig 10B–10D show the case with background interference from towers and transmission lines. The comparative analysis shows that YOLOv4-tiny has a low detection accuracy for insulator defects on high towers. As seen in Fig 10C, YOLOv5s and YOLOv6 can detect small insulator defects in complex backgrounds, but with low accuracy. The YOLOv7-tiny model struggles with detecting small defects on insulator surfaces in low-contrast images. In Fig 10A, the model failed to detect minor defects in a poorly lit environment where the insulator was broken, resulting in an exposed mandrel. However, the improved model shows better performance in identifying broken defects on insulators and can accurately detect some insulator targets even in complex backgrounds.

## 5.3 Speed comparison experiment deployed on the Jetson Nano

Hardware for the Jetson Nano: The Jetson Nano's entire design is built on the ARM architecture of the chip, which is small, inexpensive, and low power. It features a quad-core ARM CPU, a 128-core GPU, and 4GB of LPDDR4 memory.

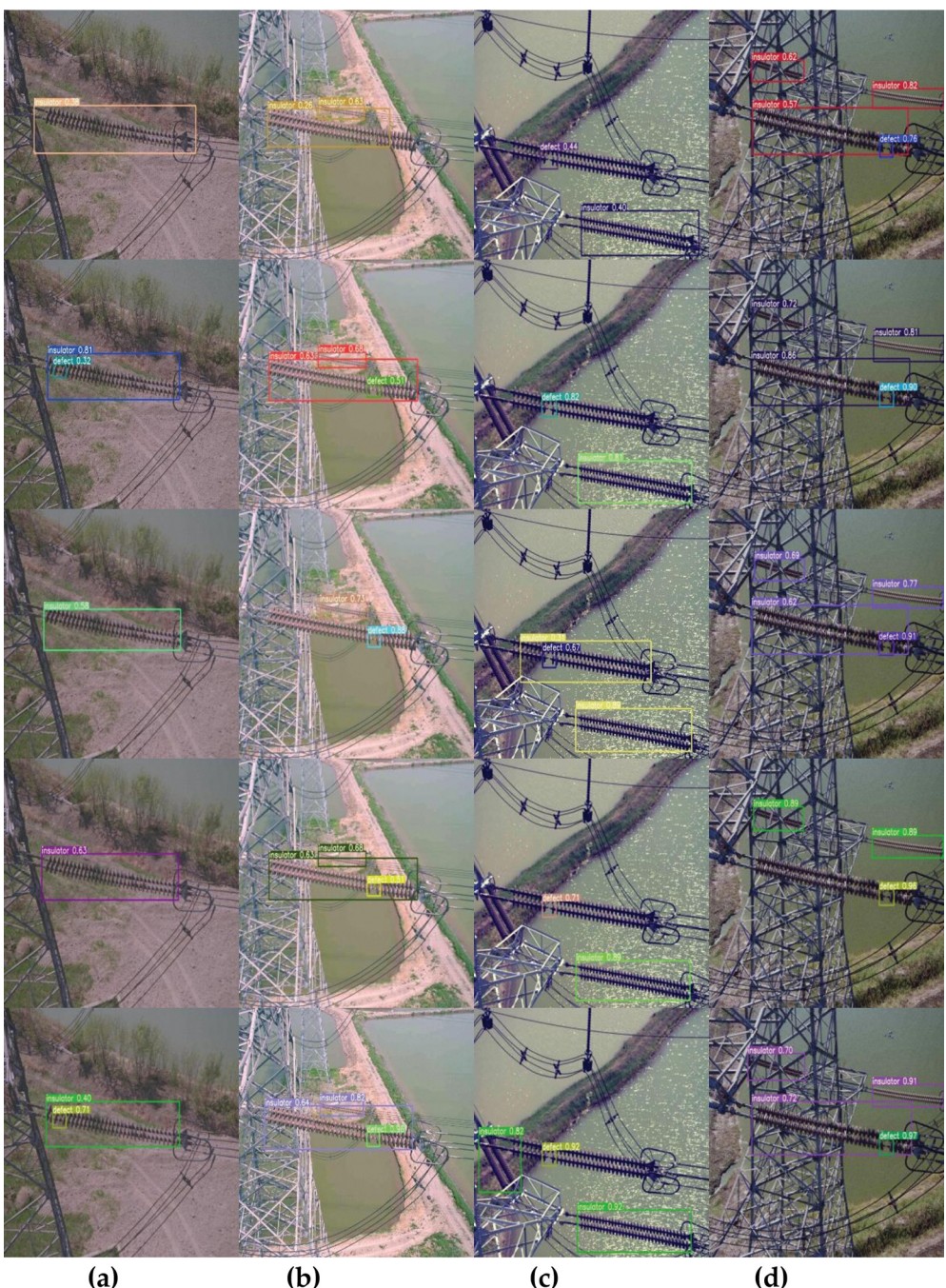

(a) (b) (c) (d)

**Fig 10. The detection results of different methods.**

Jetson Nano environment builds: YOLOv7-tiny is based on the Pytorch framework and requires Pytorch version 1.7 or higher. To meet the requirements, install JetPack SDK 4.6.1 according to the official Nvidia documentation, which provides Ubuntu 18.04, CUDA 10.2, and CUDNN 8.2, and Create a standalone runtime environment for YOLOv7 using Archi-conda, install Pytorch 1.8.0, Torchvision 0.9.0, sourced from the official NVIDIA website. The official system default swap partition Swap memory is 2G, when installing some software or

**Table 5. Comparison speed at Jetson Nano.**

| Image size | Models | Preprocessing (ms) | Inference(ms) | NMS(ms) | FP(ms) |
|---|---|---|---|---|---|
| 640×640 | YOLOv7-tiny | 7.2 | 128.2 | 37.8 | 173.2 |
| | Our model | 2.9 | 90.7 | 24.8 | 115.5 |
| 512×512 | YOLOv7-tiny | 2.9 | 79.4 | 27.8 | 110.1 |
| | Our model | 1.6 | 72.7 | 24.3 | 97.7 |
| 416×416 | YOLOv7-tiny | 1.3 | 69.5 | 28.3 | 99.1 |
| | Our model | 1.3 | 65.8 | 18.2 | 85.3 |

running larger scale computing, there is often a pop-up to remind that Swap memory is insufficient, to avoid this, use the command to increase 6G Swap memory and turn on the maximum power mode, at this time the power is 10W.

At an image resolution of 640×640, the forward transmission time of this refined model is 33.3% faster than the original model, as shown in Table 5. As the input resolution decreases, the inference speed of the model's forward transmission elapsed time gradually approaches, but this improved model is still the fastest.

## 6 Conclusion

We propose a lightweight YOLOv7 insulator defect detection algorithm in this paper, which is available for UAV inspection tasks. This algorithm can address solve problems encountered in the process of detecting defective insulators, such as inadequate target extraction, Confusion of objectives and context, and a large number of network operations that are difficult to embed in edge devices. Insulator detection differs from other generic target detection tasks in that insulators have some specific shape, scale and texture characteristics. Also, insulator detection in complex backgrounds is challenging. These task-specific features require algorithms that can better capture and utilize this feature information when processing insulator detection. Therefore, the use of techniques such as DSC, SE, GSConv, and EIoU can help improve the insulator detector's ability to sense and understand insulator-specific features and thus improve detection performance. In addition, these techniques can also reduce computational complexity to a certain extent, enabling insulator detection algorithms to be real-time and efficient in practical applications. A series of experiments were conducted to evaluate the effectiveness of the DSC-SE module in reducing network parameters in the backbone network through depth-separable convolution. The results showed that the introduction of the SE attention mechanism improved the network's extraction capability. Additionally, the GSConv was found to be effective in retaining detailed information of the input image by using SC with DSC mixing and washing operation in feature fusion network. The accuracy was ensured by using the VOVGSCSP module based on GSConv. Finally, the EIoU loss function was found to increase the model's effectiveness in fitting the real boxes to the prediction frame during training. This resulted in faster model convergence and a 0.4% increase in mAP, without affecting the number of model parameters or computational effort. The model has a high detection accuracy of 95.2% on the insulator dataset. It has a relatively low parametric number of only 3.7×106, FLOPs of 5.9G, and a decent FPS of 13 when measured on the Jetson Nano.

To address the issue of insufficient computing power and platform portability, the use of 5G communication technology in the future will be considered to offload the processing work to the cloud. This will enable inspectors to carry display devices only, achieving intelligent real-time inspection.

## Acknowledgments

The authors would like to thank the anonymous reviewers for their critical and constructive comments.

## Author Contributions

**Conceptualization:** Yulu Zhang.

**Data curation:** Jiazhao Li.

**Formal analysis:** Yulu Zhang, Gang Wang.

**Investigation:** Gang Wang.

**Methodology:** Yulu Zhang.

**Software:** Yulu Zhang.

**Supervision:** Wei Fu, Juan Ma, Gang Wang.

**Validation:** Jiazhao Li, Juan Ma.

**Visualization:** Wei Fu.

**Writing – original draft:** Yulu Zhang.

**Writing – review & editing:** Yulu Zhang, Gang Wang.

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
