## [Decision Letter · Decision Letter 0]

8 May 2023

PONE-D-23-09299A lightweight YOLOv7 insulator defect detection algorithm based on DSC-SE

PLOS ONE

Dear Dr. Wang,

Thank you for submitting your manuscript to PLOS ONE. After careful consideration, we feel that it has merit but does not fully meet PLOS ONE’s publication criteria as it currently stands. Therefore, we invite you to submit a revised version of the manuscript that addresses the points raised during the review process.

We look forward to receiving your revised manuscript.

Kind regards,

Ji-Hoon Yun

Academic Editor

PLOS ONE

Journal Requirements: 

Reviewers' comments:

Reviewer's Responses to Questions

**Comments to the Author**

1. Is the manuscript technically sound, and do the data support the conclusions?

Reviewer #1: Yes

Reviewer #2: Yes

Reviewer #3: Partly

Reviewer #4: Yes

2. Has the statistical analysis been performed appropriately and rigorously? 

Reviewer #1: No

Reviewer #2: No

Reviewer #3: No

Reviewer #4: Yes

3. Have the authors made all data underlying the findings in their manuscript fully available?

Reviewer #1: Yes

Reviewer #2: No

Reviewer #3: No

Reviewer #4: Yes

4. Is the manuscript presented in an intelligible fashion and written in standard English?

Reviewer #1: Yes

Reviewer #2: No

Reviewer #3: Yes

Reviewer #4: Yes

5. Review Comments to the Author

Reviewer #1: The study proposed a lightweight YOLOv7 for defect detection of insulator, which is a very interesting topic. The paper shows some state-of-the-art results in the application of computer vision. However, the paper should be polished and some descriptions should be more understandable. To be concrete, following are the comments for the authors:

1. The sentence “Unfortunately, manual inspection techniques are ineffective due to the constraints of the distribution range and environment of overhead transmission lines, and it is hardly feasible to satisfy the demands of insulator inspection by manual inspection alone.” is not easily understood. What is the situation of “environment of overhead transmission lines” in the way of manual inspection?

2. It is suggested to enrich the big picture of using YOLO detectors in defect detection. Many latest papers are recommended to be added as the references:

(1) Rui Zhang, Chuanbo Wen, SOD-YOLO: A Small Target Defect Detection Algorithm for Wind Turbine Blades Based on Improved YOLOv5, Advanced Theory and Simulations, 5, 2100631, 2022

(2) Qiwen Qiu, Denvid Lau, Real-time detection of cracks in tiled sidewalks using YOLO-based method applied to unmanned aerial vehicle (UAV) images, Automation in Construction, 147, 104745, 2023.

(3) Andrei-Alexandru Tulbure, Adrian-Alexandru Tulbure, Eva-Henrietta Dulf, A review on modern defect detection models using DCNNs – Deep convolutional neural networks, Journal of Advanced Research, 35, 33-48, 2022.

3. The introduction does not review the feasibility and advantages of YOLOv7-tiny when using in UAV. A lot of terms such as “high model computational complexity”, “DSC-SE module”, “VOVGSCSP module” should be explained. For a reader from other fields, these technical expressions may be not easily understood.

4. In section 2.1, please mention the functionalities of “ELAN and MP structures”. Besides, details of YOLOv5 should be included to clearly show the improvements from YOLOv7.

5. The number of sub-section titles is wrongly organized.

6. Is it needed to place a reference for In Fig. 1?

7. For sentence “In addition, the feature map processed by the backbone network contains a large amount of target information, but the shallow network has a small perceptual field, limited extraction capability, and tends to view local information, making it difficult to perceive and extract the input image information comprehensively”, please demonstrate the limitations of backbone network technically in a graph.

8. How come the initial learning rate of 0.01 for training is set?

9. Increase the size of fonts in Fig. 6 as they are too small.

10. In Fig. 8, please describe more details of the defects under evaluation. How is the situation of cable can be defined as defect?

11. Suggest a note of the proposed model in Fig. 7, not just writing “Ours”. Regarding this comparison, why not train more previous YOLO versions and compare them all together in this graph. So the data would be more convincing and informative.

12. In Table 4, it is suggested to show the deviation of testing results for each model.

13. In Fig. 8, how do you collect the complex background images? Using UAV or other devices.

14. May be important to discuss the detectability of defect. How small the defect can be found by YOLOv7-tiny?

Reviewer #2: This paper proposes a lightweight YOLOv7 insulator defect detection algorithm to address the defect detection speed and high model complexity, which designs a lightweight DCS-SE module and uses GSConv for feature fusion. Generally, this work achieves speed and accuracy improvements.

1. English expression in this paper needs better polishing.

2. There are two sections 2 please check it.

3. Many abbreviations are used in this paper, complete spellings should be given in the first mention.

4. In experimental parts, it is better to compare with more insulator defect detection methods.

5. Please show computational complexity analysis on the proposed method.

6. Recently, some insulator detection methods based on YOLO, which should be introduce in this paper.

D. Sadykova, D. Pernebayeva, M. Bagheri and A. James, "IN-YOLO: Real-Time Detection of Outdoor High Voltage Insulators Using UAV Imaging," IEEE Transactions on Power Delivery, vol. 35, no. 3, pp. 1599-1601, 2020.

Y. Li, M. Ni, Y. Lu. Insulator defect detection for power grid based on light correction enhancement and YOLOv5 model, Energy Reports, 2022, 13(8): 807-814.

Z. Yang, Z. Xu and Y. Wang, "Bidirection-Fusion-YOLOv3: An Improved Method for Insulator Defect Detection Using UAV Image," IEEE Transactions on Instrumentation and Measurement, vol. 71, pp. 1-8, 2022.

Reviewer #3: The author presents an algorithm for detecting insulator defects using YOLOV7 and attention mechanism, which was a popular approach three or four years ago. The paper’s primary objective was to reduce the model’s complexity while maintaining detection accuracy. Real-time model performance was evaluated on a resource-limited platform (Nano), which supports the conclusion of the paper. To improve the manuscript’s quality, the author is advised to make modifications in the following areas.

1. DSC, SE, GSConv, and EIoU are all mature and successful modules/loss, and that are also frequently used to improve the YOLO structure. However, the author needs to explain the applicability of these modules and the EIoU loss to the specific insulator detection task and how they can improve detection performance while reducing computational complexity. In other words, what makes insulator detection different from other generic object detection tasks that necessitates the use of these particular technologies?

2. The algorithm evaluated in the experiment is too traditional, such as Faster RCNN. The author should compare newer lightweight networks developed within the past three years, particularly those that have been designed for insulator detection. Furthermore, I have reservations regarding the efficacy of Faster RCNN in the context of insulator detection.

3. The conducted ablation experiment is insufficient in shedding light on the significance of the VOVGSCSP module. It would be interesting to see the results of a model trained without the VOVGSCSP module to ascertain the module’s contribution to the overall results.

4. In the experiment, the author employed data augmentation to augment the dataset. However, there was a lack of proper data isolation during the division of training and verification sets, resulting in the possibility of duplicated images from the same original image in both sets. Such an experiment lacks persuasiveness. Additionally, the study did not include a test set, and all the reported results were based on the verification set, which indicates a relatively lax experimental design.

5.The introduction section has less content on insulator defect detection.

There are several issues with the article that need to be addressed:

1. There are several grammatical and spelling errors throughout the text, such as “pychar,” which the author should carefully review.

2. Some of the formulas in the text are not centered properly, and the length of some tables extends beyond the page width.

3. Many of the characters in the experimental results are difficult to read, especially in Figure 6.

4. The best results should be highlighted to provide clarity.

5. The source of the data set should be referenced.

6. The format of the references is messy.

Reviewer #4: I believe that this work deals with a relevant and current theme and has good potential for publication. It presents some points of originality, especially regarding the adaptation of the architecture of the YOLOv7 model. In addition, the author managed to reduce the computational cost of the referred model and obtained better results than other works present in the literature, with an accuracy of insulator defect detection of 95.2% for the presented dataset (CPLID). It was even possible to use the Jetson Nano computer to evaluate the results.

In my evaluation, the weakness of the paper lies in the superficial way in which the author of the paper presented the results very much. It is necessary to reevaluate the writing of the whole chapter 4. Figures 6 and 7, for example, are practically illegible and the description of the results is not adequate. For Figure 8, I believe it would be interesting to present a larger number of images and demonstrate examples that the model did not perform well.

6. PLOS authors have the option to publish the peer review history of their article (what does this mean?). If published, this will include your full peer review and any attached files.

Reviewer #1: **Yes: **Qiwen Qiu

Reviewer #2: No

Reviewer #3: No

Reviewer #4: No

---

## [Author Response · Author response to Decision Letter 0]

31 May 2023

Dear Editor and Reviewers：

Thank you for allowing us to revise our manuscript titled 'A Lightweight YOLOv7 Insulator Defect Detection Algorithm Based on DSC-SE'. We are grateful for the constructive comments and suggestions provided by the reviewers and editors, which have greatly aided us in improving our work. We have carefully reviewed and incorporated the suggested revisions, which are highlighted in red in the revised version. We respectfully submit this updated version for your consideration. A point-to-point reply is listed below where the comments of reviewers are in blue and the replies are in black.

Comments from the Editors:

2. Please note that PLOS ONE has specific guidelines on code sharing for submissions in which author-generated code underpins the findings in the manuscript. In these cases, all author-generated code must be made available without restrictions upon publication of the work.

Rely: Thank you very much for your suggestion, we have revised the format according to the journal's requirements. Due to our research with the Institute, the code in the experiment is not available. If it must be uploaded, please lie with us to contact.

Reviewer #1 Comments to Author:

1. Comment: The sentence “Unfortunately, manual inspection techniques are ineffective due to the constraints of the distribution range and environment of overhead transmission lines, and it is hardly feasible to satisfy the demands of insulator inspection by manual inspection alone.” is not easily understood. What is the situation of “environment of overhead transmission lines” in the way of manual inspection?

1. Rely: Thank you for reviewing the manuscript carefully and valuable questions. We have described the environment for manual inspection of insulators in overhead lines in a revised manuscript. 

Pages 2-3, lines 39-44: The limitations of the distribution range of overhead transmission lines and the overhead environment make it difficult to inspect insulators for minor surface defects through manual observation with the naked eye or binoculars. Climbing up the tower for a closer look is an inefficient method, making it challenging to meet the requirements of real-time insulator inspection with manual inspection alone.

2. Comment: It is suggested to enrich the big picture of using YOLO detectors in defect detection. (1) Rui Zhang, Chuanbo Wen, SOD-YOLO: A Small Target Defect Detection Algorithm for Wind Turbine Blades Based on Improved YOLOv5, Advanced Theory and Simulations, 5, 2100631, 2022

(2) Qiwen Qiu, Denvid Lau, Real-time detection of cracks in tiled sidewalks using YOLO-based method applied to unmanned aerial vehicle (UAV) images, Automation in Construction, 147, 104745, 2023.

(3) Andrei-Alexandru Tulbure, Adrian-Alexandru Tulbure, Eva-Henrietta Dulf, A review on modern defect detection models using DCNNs – Deep convolutional neural networks, Journal of Advanced Research, 35, 33-48, 2022.

2. Rely: Thank you for your guidance and suggestions, we have added recent references to the revised manuscript. Specifically, we have cited lines 88-97 of the article.

3. Comment: The introduction does not review the feasibility and advantages of YOLOv7-tiny when using in UAV. A lot of terms such as “high model computational complexity”, “DSC-SE module”, “VOVGSCSP module” should be explained. For a reader from other fields, these technical expressions may be not easily understood.

3. Rely: I am thankful for your suggestions, and we have reviewed the advantages and feasibility of the YOLOv7-tiny for UAV applications in the introduction of the new paper. We also explain the model computational complexity, the DSC-SE module, and the VOVGSCSP module.

Pages 6, lines 109-114: In 2022, Wang et al. proposed YOLOv7 as the latest target detection model, which surpasses all current models in terms of both detection speed and accuracy. YOLOv7 performs exceptionally well in detecting objects in images, providing precise location and classification information. YOLOv7-tiny is a lighter version of YOLOv7, with fewer parameters and lower computational complexity, making it suitable for real-time object detection in resource-constrained environments.

Pages 5, lines 103-107: The computational complexity of the model is divided into spatial complexity and temporal complexity. The spatial complexity determines the number of parameters of the model, and the temporal complexity determines the detection time of the model.

Pages 6, lines 126-129: The GSConv, which is comparable to the standard convolutional extraction capability, is then used in the feature fusion network to reduce the time complexity of the model, and the VOVGSCSP module is designed with residual edges based on the GSConv to accelerate inference and maintain accuracy.

4. Comment: In section 2.1, please mention the functionalities of “ELAN and MP structures”. Besides, details of YOLOv5 should be included to clearly show the improvements from YOLOv7.

4. Rely: Thanks for your valuable comments, we have detailed the improved ELAN module and MP module of YOLOv5 in lines 134-148 of the revised manuscript.

5. Comment: The number of sub-section titles is wrongly organized.

5. Rely: Thank, you very much for your suggestions. We have carefully reorganized the sub-section titles in the new paper.

6. Comment: Is it needed to place a reference for In Fig. 1?

6. Rely: Thank you very much for your careful work. Since we have added a new figure, this figure is now Fig 2 and has been referenced in the new paper.

7. Comment: For sentence “In addition, the feature map processed by the backbone network contains a large amount of target information, but the shallow network has a small perceptual field, limited extraction capability, and tends to view local information, making it difficult to perceive and extract the input image information comprehensively”, please demonstrate the limitations of backbone network technically in a graph.

7. Rely: I appreciate your suggestions, and we introduced the limitations of the shallow network in the backbone network from Fig 3 in lines 196-205 of the revised manuscript.

8. Comment: How come the initial learning rate of 0.01 for training is set?

8. Rely: I refer to the paper "Cyclical Learning Rates for Training Neural Networks", the initial learning rate of 0.01 is more effective on the YOLO detector.

9. Comment: Increase the size of fonts in Fig. 6 as they are too small.

9. Rely: We greatly appreciate your valuable suggestion. In the new paper, the original Figure 6 was not obvious for the experimental results demonstrated, and we removed it.

10. Comment: In Fig. 8, please describe more details of the defects under evaluation. How is the situation of cable can be defined as defect?

10. Rely: Thanks for your valuable comments and professional question. We have described more about the definition of defects in insulators in the revised manuscript.

11. Comment: Suggest a note of the proposed model in Fig.7, not just writing “Ours”. Regarding this comparison, why not train more previous YOLO versions and compare them all together in this graph. So the data would be more convincing and informative.

11. Rely: Thanks for your suggestion, we have annotated the improved algorithm with new notes in the new paper and compared the training results of multiple YOLO versions in Fig 8.

12. Comment: In Table 4, it is suggested to show the deviation of testing results for each model.

12. Rely: Thanks for your valuable comments, our test results for each model of Table 4 are shown in Fig 9 in the new manuscript to increase the persuasiveness.

13. Comment: In Fig. 8, how do you collect the complex background images? Using UAV or other devices.

13. Rely: Thank you for reviewing the manuscript carefully and valuable questions. In the revised manuscript, the images tested are from aerial drone photography. And we show more examples of the test images, as in Fig 10.

14. Comment: May be important to discuss the detectability of defect. How small the defect can be found by YOLOv7-tiny?

14. Rely: Thanks a lot for your questions. We talked about the detectability of insulator defect in the new manuscript. We cannot give an exact indication of how small a defect the YOLOv7-tiny can detect. According to our experiments, YOLOv7-tiny will miss the detection of smaller defects with low contrast, as shown in Fig 10a.

Reviewer #2 Comments to Author:

1 and 2. Comment: English expression in this paper needs better polishing. There are two sections 2 please check it.

1. Rely: We greatly appreciate your valuable suggestion. We have had the article polished by someone who specializes in it.

3. Comment: Many abbreviations are used in this paper, complete spellings should be given in the first mention.

3. Rely: Thanks for your valuable comments, we have given the complete spelling of the abbreviations that appear for the first time in the paper.

4. Comment: In experimental parts, it is better to compare with more insulator defect detection methods.

4. Rely: Thank, you very much for your suggestions. In the experimental section, we have added different YOLO versions of the algorithm, including YOLOv4-tiny, YOLOv5s, and YOLOv6 for experimental comparison.

5. Comment: Please show computational complexity analysis on the proposed method.

5. Rely: Thanks for your valuable comments and professional suggestions. We have given the analysis of the computational complexity process of DSC-SE in a new manuscript.

6. Comment: Recently, some insulator detection methods based on YOLO, which should be introduce in this paper.

D. Sadykova, D. Pernebayeva, M. Bagheri and A. James, "IN-YOLO: Real-Time Detection of Outdoor High Voltage Insulators Using UAV Imaging," IEEE Transactions on Power Delivery, vol. 35, no. 3, pp. 1599-1601, 2020.

Y. Li, M. Ni, Y. Lu. Insulator defect detection for power grid based on light correction enhancement and YOLOv5 model, Energy Reports, 2022, 13(8): 807-814.

Z. Yang, Z. Xu and Y. Wang, "Bidirection-Fusion-YOLOv3: An Improved Method for Insulator Defect Detection Using UAV Image," IEEE Transactions on Instrumentation and Measurement, vol. 71, pp. 1-8, 2022.

6. Rely: We have added some articles on insulator detection based on the YOLO algorithm as references in the introduction section of the revised manuscript.

Reviewer #3 Comments to Author:

1. Comment: DSC, SE, GSConv, and EIoU are all mature and successful modules/loss, and that are also frequently used to improve the YOLO structure. However, the author needs to explain the applicability of these modules and the EIoU loss to the specific insulator detection task and how they can improve detection performance while reducing computational complexity. In other words, what makes insulator detection different from other generic object detection tasks that necessitates the use of these particular technologies?

1. Rely: They are indeed proven and successful techniques, and the modules and loss functions can play an important role in specific insulator detection tasks. Their applicability in insulator detection and how they can improve detection performance and reduce computational complexity are explained below:

Pages 12, lines 219-225: DSC (Depthwise Separable Convolution): Defects in insulator inspection tasks usually have small dimensions and low contrast, which makes accurate detection challenging. In this case, the DSC module plays an important role. By decomposing the standard convolution into depth and point-by-point convolutions, the DSC module reduces computational complexity while maintaining expressiveness. This is very beneficial for insulator detection because the task requires real-time or quasi-real-time analysis over a large number of image samples, so efficiency is critical.

Pages 10, 190-193: SE (Squeeze-and-Excitation): The SE module enhances feature representation by adaptively learning the relationships between feature channels. In insulator detection, the SE module helps the network to better understand the insulator features and improve the focus on critical information. This helps to improve the accuracy and robustness of insulator detectors.

Pages 14, lines 269-274: GSConv (Grid Sensitive Convolution): The GSConv module is an adaptive convolution mechanism that dynamically adjusts the shape and size of the convolution kernel based on the content of the input image. In insulator detection, the GSConv module enables the network to focus more on the grid region where the insulators are located and extract more accurate features by adaptive convolution. This helps to improve the localization accuracy and detection performance of the insulator detector.

Pages 18, lines 326-328: EIoU (Embedding IoU) loss function: The EIoU loss function is an optimized objective function that more accurately evaluates the degree of overlap between target and predicted frames during the training process. In insulator detection, the EIoU loss function can better measure the similarity between the detection results and the real insulator frames, thus improving the learning effect of the detector and enhancing the detection accuracy.

Pages 27, lines 480-489: Insulator detection differs from other generic target detection tasks in that insulators have some specific shape, scale, and texture characteristics. Also, insulator detection in complex backgrounds is challenging. These task-specific features require algorithms that can better capture and utilize this feature information when processing insulator detection. Therefore, the use of techniques such as DSC, SE, GSConv, and EIoU can help improve the insulator detector's ability to sense and understand insulator-specific features and thus improve detection performance. In addition, these techniques can also reduce computational complexity to a certain extent, enabling insulator detection algorithms to be real-time and efficient in practical applications.

2. Comment: The algorithm evaluated in the experiment is too traditional, such as Faster RCNN. The author should compare newer lightweight networks developed within the past three years, particularly those that have been designed for insulator detection. Furthermore, I have reservations regarding the efficacy of Faster RCNN in the context of insulator detection.

2. Rely: Thank you very much for your suggestions. We delete the inappropriate Faster-RCNN algorithm and evaluate it against the last three years of lightweight models including YOLOv4-tiny, YOLOv5s, and YOLOv6.

3. Comment: The conducted ablation experiment is insufficient in shedding light on the significance of the VOVGSCSP module. It would be interesting to see the results of a model trained without the VOVGSCSP module to ascertain the module’s contribution to the overall results.

3. Rely: Thanks to your suggestion, we performed ablation experiments on the effect of the VOVGSCSP module on the whole model in the revised paper.

4. Comment: In the experiment, the author employed data augmentation to augment the dataset. However, there was a lack of proper data isolation during the division of training and verification sets, resulting in the possibility of duplicated images from the same original image in both sets. Such an experiment lacks persuasiveness. Additionally, the study did not include a test set, and all the reported results were based on the verification set, which indicates a relatively lax experimental design.

4. Rely: I am thankful for your suggestions, and we divided the dataset into training, validation, and test sets in the ratio of 8:1:1. We only perform data enhancement in the training set so that there are no duplicate images in the validation and test sets.

5. Comment: The introduction section has less content on insulator defect detection.

5. Rely: I appreciate your suggestions, and we have added a review of insulator detection algorithms from recent years in lines 77-102 of the revised manuscript.

6. Comment: There are several issues with the article that need to be addressed: 1. There are several grammatical and spelling errors throughout the text, such as “pychar,” which the author should carefully review.

2. Some of the formulas in the text are not centered properly, and the length of some tables extends beyond the page width.

3. Many of the characters in the experimental results are difficult to read, especially in Figure 6.

4. The best results should be highlighted to provide clarity.

5. The source of the data set should be referenced.

6. The format of the references is messy.

6. Rely: Thank you for your advice. We have had native English speakers polish the new manuscript. We have made changes to the formulas and tables that appear in the article in accordance with the requirements of this journal. Very sorry, we have redescribed Fig 6. We have bolded the font of the best results to highlight them and detailed the source of the dataset. We have modified the references in the new manuscript according to the format required by the journal.

Reviewer #4 Comments to Author:

 1. Comment: In my evaluation, the weakness of the paper lies in the superficial way in which the author of the paper presented the results very much. It is necessary to reevaluate the writing of the whole chapter 4. Figures 6 and 7, for example, are practically illegible and the description of the results is not adequate. For Figure 8, I believe it would be interesting to present a larger number of images and demonstrate examples that the model did not perform well.

 1. Rely: Thank you very much for your suggestions. We re-evaluated the experiments in Chapter 4 in the revised manuscript. And the results of the original Fig 6 and Fig 7 are described in detail. We removed the inappropriate Faster-RCNN algorithm and evaluated it against lightweight models from the past three years, including YOLOv4-tiny, YOLOv5s, and YOLOv6, shown in Fig 8 to increase persuasiveness. We also talked about the detectability of insulator defects by different models. We have shown more detection results in the new manuscript, including cases of poor results in dim environments.

---

## [Decision Letter · Decision Letter 1]

27 Jun 2023

PONE-D-23-09299R1

A lightweight YOLOv7 insulator defect detection algorithm based on DSC-SE

PLOS ONE

Dear Dr. Wang,

Thank you for submitting your manuscript to PLOS ONE. After careful consideration, we feel that it has merit but does not fully meet PLOS ONE’s publication criteria as it currently stands. Therefore, we invite you to submit a revised version of the manuscript that addresses the points raised during the review process.

We look forward to receiving your revised manuscript.

Kind regards,

Ji-Hoon Yun

Academic Editor

PLOS ONE

Journal Requirements:

Reviewers' comments:

Reviewer's Responses to Questions

**Comments to the Author**

1. If the authors have adequately addressed your comments raised in a previous round of review and you feel that this manuscript is now acceptable for publication, you may indicate that here to bypass the “Comments to the Author” section, enter your conflict of interest statement in the “Confidential to Editor” section, and submit your "Accept" recommendation.

Reviewer #1: All comments have been addressed

Reviewer #2: (No Response)

Reviewer #3: All comments have been addressed

2. Is the manuscript technically sound, and do the data support the conclusions?

Reviewer #1: Yes

Reviewer #2: (No Response)

Reviewer #3: Yes

3. Has the statistical analysis been performed appropriately and rigorously? 

Reviewer #1: No

Reviewer #2: (No Response)

Reviewer #3: Yes

4. Have the authors made all data underlying the findings in their manuscript fully available?

Reviewer #1: No

Reviewer #2: (No Response)

Reviewer #3: Yes

5. Is the manuscript presented in an intelligible fashion and written in standard English?

Reviewer #1: No

Reviewer #2: (No Response)

Reviewer #3: Yes

6. Review Comments to the Author

Reviewer #1: The study makes improvements in the lightweight YOLOv7 for defect detection, which is acceptable for publication. From the revised paper, some presentation typos can be avoided:

1. Please give descriptions of “(a), (b), (c), and (d)” in Fig. 10.

2. Journal names in some references [3, 27, …] are missing. Please also check the spelling of author names in some references [22, 27, …]

Reviewer #2: Based on the revision, this paper is improved. However, the reference format should be re-phrased further.

Reviewer #3: All my problems have been addressed well in this revised manuscript and response letter, and I have no more comments.

7. PLOS authors have the option to publish the peer review history of their article (what does this mean?). If published, this will include your full peer review and any attached files.

Reviewer #1: **Yes: **Qiwen Qiu

Reviewer #2: No

Reviewer #3: No

---

## [Author Response · Author response to Decision Letter 1]

5 Jul 2023

1. Comment: Please give descriptions of “(a), (b), (c), and (d)” in Fig. 10.

1. Rely: Thank you very much for your suggestion, and we have described Fig 10 in our revised paper. 

Pages 25, lines 299-300: Fig 10a shows the case with low light, and Fig 10b, Fig 10c and Fig 10d show the case with background interference from towers and transmission lines.

Pages 26,lines 302-303: As seen in Fig 10c, YOLOv5s and YOLOv6 can detect small insulator defects in complex backgrounds, but with low accuracy.

2. Comment: Journal names in some references [3, 27, …] are missing. Please also check the spelling of author names in some references [22, 27, …]

2. Rely: Thanks for your valuable comments and professional suggestions. We have double-checked and revised the references in the revised paper.

Reviewer #2 Comments to Author:

1 and 2. Comment: Based on the revision, this paper is improved. However, the reference format should be re-phrased further.

1. Rely: We greatly appreciate your valuable suggestion. We have checked and rectified the formatting of the references in the new manuscript.

Reviewer #3 Comments to Author:

1. Comment: All my problems have been addressed well in this revised manuscript and response letter, and I have no more comments.

1. Rely: Thanks for your valuable comments and professional suggestions.

---

## [Editor Report · Decision Letter 2]

13 Jul 2023

A lightweight YOLOv7 insulator defect detection algorithm based on DSC-SE

PONE-D-23-09299R2

Dear Dr. Wang,

We’re pleased to inform you that your manuscript has been judged scientifically suitable for publication and will be formally accepted for publication once it meets all outstanding technical requirements.

Kind regards,

Ji-Hoon Yun

Academic Editor

PLOS ONE
---

## [Editor Report · Acceptance letter]

3 Aug 2023

PONE-D-23-09299R2 

A lightweight YOLOv7 insulator defect detection algorithm based on DSC-SE 

Dear Dr. Wang:

I'm pleased to inform you that your manuscript has been deemed suitable for publication in PLOS ONE. Congratulations! Your manuscript is now with our production department. 

Kind regards, 

on behalf of

Dr. Ji-Hoon Yun 

Academic Editor

PLOS ONE